Developing a machine learning model to identify protein–protein interaction hotspots to facilitate drug discovery

Nandakumar Rohit rnandaku@asu.edu
Dinu Valentin
Program of Biomedical Informatics, College of Health Solutions, Arizona State University , Tempe, AZ , USA
Gomez Shawn
Electronic publication date: 2020 Dec 7
Publication date: 2020
Volume: 8
Electronic Location ID: e10381
Received 2020 May 20; Accepted 2020 Oct 27
Copyright: © 2020 Nandakumar and Dinu
Copyright year: 2020
Copyright holder: Nandakumar and Dinu
License: This is an open access article distributed under the terms of the Creative Commons Attribution License, which permits unrestricted use, distribution, reproduction and adaptation in any medium and for any purpose provided that it is properly attributed. For attribution, the original author(s), title, publication source (PeerJ) and either DOI or URL of the article must be cited.
License URL: https://creativecommons.org/licenses/by/4.0/

Keywords: Machine learning, Protein-protein interaction, Drug discovery

Funding: The authors received no funding for this work.

==============================
Throughout the history of drug discovery, an enzymatic-based approach for identifying new drug molecules has been primarily utilized. Recently, protein–protein interfaces that can be disrupted to identify small molecules that could be viable targets for certain diseases, such as cancer and the human immunodeficiency virus, have been identified. Existing studies computationally identify hotspots on these interfaces, with most models attaining accuracies of ~70%. Many studies do not effectively integrate information relating to amino acid chains and other structural information relating to the complex. Herein, (1) a machine learning model has been created and (2) its ability to integrate multiple features, such as those associated with amino-acid chains, has been evaluated to enhance the ability to predict protein–protein interface hotspots. Virtual drug screening analysis of a set of hotspots determined on the EphB2-ephrinB2 complex has also been performed. The predictive capabilities of this model offer an AUROC of 0.842, sensitivity/recall of 0.833, and specificity of 0.850. Virtual screening of a set of hotspots identified by the machine learning model developed in this study has identified potential medications to treat diseases caused by the overexpression of the EphB2-ephrinB2 complex, including prostate, gastric, colorectal and melanoma cancers which are linked to EphB2 mutations. The efficacy of this model has been demonstrated through its successful ability to predict drug-disease associations previously identified in literature, including cimetidine, idarubicin, pralatrexate for these conditions. In addition, nadolol, a beta blocker, has also been identified in this study to bind to the EphB2-ephrinB2 complex, and the possibility of this drug treating multiple cancers is still relatively unexplored.

Introduction

Drug discovery is the scientific process where new drugs and small molecules are developed and identified to treat certain conditions. Throughout most of the history of drug discovery, an enzymatic-based (lock and key) approach for identifying new drug molecules was utilized (Bakail & Ochsenbein, 2016). As a result, many drugs targeting G-protein coupled receptors (GPCRs), which interact via this approach, constitute about 34% of the drugs in the market today (Hauser et al., 2017).

Protein-protein interfaces have been of particular interest in regards to drug discovery, such as the EphA4-EphrinB2 complex, which is considered to be conformationally flexible (Ma & Nussinov, 2014). Protein-protein interfaces can be stabilized or disrupted to identify small molecules that could be viable targets for certain diseases such as cancer and the human immunodeficiency virus (HIV). Identifying residue hotspots on these protein–protein interfaces and repurposing existing drugs to target these new hotspots can lead to novel drug targets, ultimately leading to new therapeutic treatments (Scott et al., 2016). Although protein-based drug discovery (as opposed to enzymatic-based drug discovery) is a relatively new and emerging field, recent studies have shown promising results in regard to its potential in a wide range of fields from drug discovery to drug repositioning. For example, the SpotOn study has produced remarkable results in regards to identifying hotspots that are viable for drug discovery, and AnchorQuery, which identifies small molecule protein-interaction inhibitors (Moreira et al., 2017; Koes, Dömling & Camacho, 2018).

In addition, PPI-based peptide drug discovery has been used to identify new therapeutic targets by disrupting PPIs. Major advances in docking simulations and models in recent years have yielded to be effective in more accurately identifying peptide-protein interactions. Although peptide-based PPI drug discovery does have its challenges, such as limited bioavailability and solubility of peptides, this emerging field highlights potentially exciting advances in computationally aided protein–protein interaction based discovery techniques with the use of interfering peptides (Lee et al., 2019).

Currently, only 10–14% of the human proteome is considered to be “druggable”, and most targets with published leads are in the rhodopsin-like GPCR family, with a smaller number in cation channels and protein kinases (Hopkins & Groom, 2002; López-Cortés et al., 2019). Druggability is the ability for a drug to bind to a specific target. As protein-based drug discovery is a relatively new field compared to traditional drug discovery, more research is needed to identify new hotspots on protein–protein interfaces. Existing studies do computationally identify hotspots on these interfaces, but most of the models developed only attain accuracies of around 70% (Kim, Chivian & Baker, 2004; Tuncbag, Keskin & Gursoy, 2010). Moreover, many studies do not effectively integrate information relating to amino acid chains and other structural information relating to the complex/interface, and/or have completely different approaches to predict the likelihood of hotspots on a particular interface.

For example, molecular dynamics (MD) simulations have been used to elucidate the mechanisms of protein interactions and their viability for drug discovery. This strategy has mixed results however—although the approach of molecular dynamics simulations have relatively high predictive power, these simulations are computationally expensive (Cukuroglu et al., 2014). In contrast, knowledge-based machine learning techniques have the advantage of providing accurate results based on the properties/features of a specific interaction. Machine learning and other statistical approaches allow for a high predictive power of hotspot detection, while being computationally efficient, provided that the features inputted into the model are relevant.

This leads to the proposed research question: “Can the development of a machine learning model lead to the discovery of new druggable targets and new drug-disease associations?” The hypothesis was that the integration of different protein–protein interaction features will lead to promising new hotspots. In addition, new drug-disease associations could potentially be identified from these hotspots to treat deadly diseases such as cancer.

To test this hypothesis, (1) a machine learning model was developed and (2) its ability to integrate multiple features, including structural information, such as that associated with amino-acid chains, to enhance the ability to predict protein–protein interface hotspots was evaluated. In addition, virtual drug screening of a set of hotspots identified by the machine learning model developed herein was performed in order to identify potentially new drug-disease associations. Phase 1 consisted of developing the machine learning model to identify potential protein–protein interface hotspots that could be viable as a drug target, using the cancer-associated EphB2-ephrinB2 protein complex (PDB code: 1KGY) for illustration. Phase 2 of this project aimed to identify small molecules that could act as inhibitors or disruptors to the hotspots identified for further analysis in Phase 1.

The machine learning model developed in Phase 1 achieved an area under receiver operating characteristic (AUROC) of 0.842 on the testing set, and identified residues 1122–1126 on this complex as potential hotspot residues. This information was then used to generate a pharmacophore in Phase 2 which identified nine drug candidates to disrupt the EphB2-ephrinB2 complex. Out of these candidates, further literature review identified four drug candidates that could treat diseases that are overexpressed by this complex: cimetidine, idarubicin, pralatrexate, and nadolol. Although nadolol has been relatively unexplored in its potential of treating certain cancers, a drug with a similar chemical makeup, propranolol, has been identified to treat multiple cancers including colon cancer, which is linked to the overexpression of the EphB2-ephrinB2 complex (Pantziarka et al., 2016; Işeri et al., 2014), and thus highlights significant repositioning opportunities for nadolol.

Methods

Dataset collection and feature aggregation

As a starting point, the dataset and codebase from the SpotOn study (Moreira et al., 2017) were acquired. This study was selected as the starting point for its high effectiveness in identifying potential hotspots that could aid in drug discovery. The SpotOn database already has information regarding amino acid composition, solvent-accessible surface area (SASA) information, position-specific scoring matrices, the number of amino acids at 2.5 and 4.0 Angstrom, the number of nearby hydrophobic residues, the total change in SASA, the number of interfacial residues, pseudo-amino acid composition, and scales-based descriptors of 2D and 3D descriptors from the protr R package (see below) for a total of 881 features.

In order to add more information to this dataset to better aid model prediction, the protr R package (Xiao et al., 2015) was used to add more features related to amino acid composition, dipeptide composition, etc., to the already pre-existing data. Additionally, data related to pair potential, complex/monomer accessible surface area, residue information, amino acid information, etc. were extracted from the HotPoint database (Tuncbag, Keskin & Gursoy, 2010) and then added to the pre-existing dataset. This data was added to add more information regarding the entire protein complex, as evidenced by most of protr’s features, and to add residue specific features such as pair potential that could improve predictive power. The addition of new features in the protr R package and the HotPoint database led to a total of 2,323 features.

Upon further investigation of the SpotOn dataset, we found that chains I of proteins with PDB code 2FTL, 3SG8, and 1CH0 do not exist as specified in the Protein Data Bank. In the SpotOn study, these chains are specified, and features were derived for these chains; however, in this study, as additional features are added and these chains could not be identified, these chains have been removed from our dataset. This leads to a total of 520 protein residues, lower than SpotOn’s 534 protein residues. A total of 398 residues are labeled as non-hotspots, and 122 residues are labeled as hotspots.

In order to derive features on our prediction dataset with the EphB2-ephrinB2 complex (PDB code: 1KGY), we first downloaded the structure from the Protein Data Bank, and ran this structure through the SpotOn’s codebase/pipeline to collect features specific to the SpotOn study. Then, we sequentially added additional features unique to this study, such as from the protr’s R package and features from the HotPoint database. Missing values are assigned the average value of all non-missing values in a feature.

Preprocessing and feature engineering

Similar to the SpotOn study, both the training and testing sets were normalized, and the testing set was normalized using mean and standard deviation of the training set. In addition, before the model was run, data balance had to be accounted for, and oversampling was performed in order to retain the properties of the majority class without sacrificing the information available in this class (More, 2016). SMOTE, or synthetic minority oversampling technique, was performed with k = 5 nearest neighbors (Chawla et al., 2002). To account for multicollinearity, principal component analysis was also performed. This leads to four different combinations: a pipeline without any changes to the training data, a pipeline with only SMOTE applied, a pipeline with only PCA applied, and a pipeline with both SMOTE and PCA applied.

Before the model was trained, the dataset was first subjected to feature engineering. Three existing features that were selected for further exploration are the number of intermolecular contacts within 4.0 Angstroms (#Dist-4.0), the number of hydrophobic contacts (#Hydrophobic), and the pair potential of a specific residue (Pair Potential). We hypothesized that an increase of hydrophobic contacts would cause a decrease in hydrophobic pair potential due to the attractive interaction because of the hydrophobic effect (Israelachvili & Pashley, 1982). As a result, we multiplied both variables and multiplied by −1 to amplify the effects of this association and accounting for the inverse correlation. In addition, we hypothesized that the number of intermolecular contacts will increase the pair potential as this may lead to many body potentials, which are mostly repulsive at short distances (Byggmästar, Granberg & Nordlund, 2018). To model this association, #Dist-4.0 and #Hydrophobic are multiplied to amplify the effects as well. These two new engineered variables were named #Dist-4.0 * Pair Potential and -#Hydrophobic * Pair Potential. This lead to a total of 2,323 features on the training and testing datasets, as well as our dataset containing residue information on the crystal structure of the EphB2-ephrinB2 complex (PDB code: 1KGY).

Machine learning model selection

Five different machine learning models were selected in order to evaluate and develop a model: Logistic regression (LR), XGBoost (XGB), a balanced random forest classifier (RF), K Nearest Neighbors (KNN), multilayer perceptron neural network (MLP), and a Gaussian Naïve Bayes (GNB). This data was then split into a training:testing set ratio of 80:20. 10-fold cross validation was performed 10 times on the training set to prevent overfitting. GridSearch was performed in order to identify the best combination of hyperparameters/parameters that could yield the best results. The following hyperparameters/parameters were tested: LR, with C equal to 0.01, 0.1, 1, 10, 50, 100, 500, 1,000, and 5,000; RF with balanced class weight, with the number of estimators equal to 50, 100, 150, 250, 350, 500, and maximum depth of 5, 7, 9, 11; XGB, with a learning rate of 0.001, 0.01, 0.1, the number of estimators as 50, 100, 150, 200, and maximum depth of 4, 5, 6; KNN, with n neighbors of 1, 3, 5, 10, 15, 20; a multilayer perceptron model of hidden_layer_sizes (10, 10, 10), (50, 1), (10, 10), (10, 1), (5, 5), (5, 5, 5) and alpha of 0.0001, 0.0002, 0.0005, 0.001; and GNB with variance smoothing of 1e−8, 1e−7, 1e−6, 1e−5, 1e−4, 1e−3, and 1e−2. The metric used to identify the best model from these sets of parameters on the validation set is AUROC (area under receiver operating characteristic), and this was chosen to compare our models more accurately with the AUROC and receiver-operating characteristic curves provided in SpotOn. Four different run conditions on the four different pipelines was also run and the results are compared. The model(s) with the best run conditions on the highest scoring pre-processing dataset will be used to build an ensemble model, similar to the SpotOn study. If the ensemble model has a higher predictive capability than any individual model, the ensemble model will then be used to predict hotspots on the EphB2-ephrinB2 complex, as this complex has been overexpressed in many cancer cells, most notably in prostate, gastric, colorectal and melanoma cancers (Pasquale, 2010). PyMol was utilized to visualize the hotspots predicted on the EphB2-ephrinB2 complex.

Small molecule selection

A cluster of hotspots was identified and LigandScout (Wolber & Langer, 2005) was used to create an apo-site pharmacophore. Virtual screening was then performed on this pharmacophore to identify possible new drug indications. To perform the drug screening, an approved Drugbank (Wishart et al., 2008) database that has a library of all molecules that have molecular weight from 150 to 500 daltons was used. These small molecules were then ranked by the LigandScout software to identify molecules that most strongly conform to the pharmacophore based on the chemical and structural properties of that molecule. The drug-disease associations were then verified with scientific literature to assess the validity and efficacy of the model, and then we identified new drug-disease associations that have not been previously identified by cross-referencing existing scientific literature.

Metric calculation

In context, sensitivity is the ability for the model to identify the hotspots and the specificity/recall is the ability for the model to identify the non-hotspots, and both of these statistics are defined as: Recall=Sensitivity=TruePositiveTruePositive+FalseNegative

Specificity=TrueNegativeTrueNegative+FalsePositive

Precision is defined as: Precision=TruePositiveTruePositive+FalsePositive

F1, MCC, Kappa, and Precision-Recall are all metrics that are robust in dealing with data imbalance. They are defined as: F1=2∗precision∗recallprecision+recall

MCC=TruePositive∗TrueNegative−FalsePositive∗FalseNegative(TruePos+FalsePos)(TruePos+FalseNeg)(TrueNeg+FalsePos)(TrueNeg+FalseNeg)

Kappa=(po−pe)/(1−pe) where po is the probability of agreementassigned to any sample, and pe is the expected/hypothetical probability of chance agreement.

Precision-Recall =∑n⁡(Rn−Rn−1)Pn where Pn and Rn are precision and recall, respectively, at the nth threshold.

All of these calculations are calculated using the Scikit-learn package in Python.

Results

Phase 1

The average test metrics of each of the six algorithms tested on the four different pre-processing pipelines are shown in Table 1. As the preprocessing pipeline where SMOTE and PCA are applied has the highest AUROC, the top algorithms from this pipeline are used to create an ensemble model.

Table 1 Average test metrics of algorithms tested on pre-processing pipelines.

The “AUROC” column is bolded as to highlight the metric used to select the best pre-processing pipeline. The “SMOTE, PCA” pipeline is the best performing pipeline based on AUROC.

Test	AUROC	Accuracy	Precision	Recall/Sensitivity	F1	Precision-Recall	MCC	Kappa	Specificity	
ONLY SMOTE	0.745	0.772	0.516	0.694	0.584	0.430	0.451	0.435	0.796	
RAW	0.722	0.777	0.536	0.618	0.563	0.416	0.427	0.418	0.825	
NO SMOTE, PCA	0.702	0.785	0.535	0.549	0.539	0.403	0.402	0.400	0.856	
SMOTE, PCA	0.774	0.798	0.551	0.729	0.625	0.467	0.502	0.491	0.819	

In Table 2 are the best individual algorithms tested in the SMOTE and PCA pipeline. The best set of hyperparameters were selected using GridSearch as follows: the logistic regression with C = 1,000, the random forest classifier with maximum depth of 5 trees and the total number of estimators at 50 trees, an XGBoost classifier with learning rate 0.1, maximum depth of 5, and 50 estimators, K-nearest neighbors with 5 neighbors, a multi-layer perceptron classifier with alpha as 0.0002 and three layers of 10 neurons each, and a Gaussian Naïve Bayes of variable smoothing of 1e−5. The best performing algorithm in this pipeline is the logistic regression, as it has the highest AUROC.

Table 2 Best Individual Algorithms in SMOTE and PCA pipeline.

The “AUROC” column is bolded as to highlight the metric used to select the best individual algorithm. The logistic regression algorithm is the best performing model based on AUROC.

	AUROC	Accuracy	Precision	Recall/Sensitivity	F1	Precision-Recall	MCC	Kappa	Specificity	
LR	0.842	0.846	0.625	0.833	0.714	0.559	0.624	0.612	0.850	
RF	0.756	0.827	0.625	0.625	0.625	0.477	0.513	0.513	0.888	
GBC	0.748	0.769	0.500	0.708	0.586	0.422	0.445	0.433	0.788	
KNN	0.706	0.750	0.469	0.625	0.536	0.380	0.377	0.369	0.788	
MLP	0.829	0.827	0.588	0.833	0.690	0.529	0.591	0.575	0.825	
Gaussian	0.756	0.827	0.625	0.625	0.625	0.477	0.513	0.513	0.888	

The results of the Logistic Regression from the SMOTE and PCA pipeline were compared with SpotOn’s highest performing algorithm from the upsampling pre-processing procedure, as shown in Table 3. In order to more aptly analyze the predictive capabilities of our top performing algorithm, we adjusted the threshold to achieve a ≥0.88 specificity, as that is the specificity of the highest performing individual algorithm in SpotOn, and comparing other metrics based on this threshold change. This reduced the AUROC from 0.842, as identified in Table 2, to 0.840, as identified in Table 3, but was still greater than any individual model in SpotOn. Our algorithms perform better to that of SpotOn’s ScaledUp processing step in all six metrics as demonstrated in Table 3.

Table 3 Comparison of our study vs SpotOn.

The “LG from PCA and SMOTE” column is selected to highlight the metrics from the best performing algorithm from the best performing pipeline from our study.

Test	LG from PCA and SMOTE	SpotOn’s ScaledUp*	
Accuracy	0.865	0.79	
F1	0.731	0.52	
AUROC	0.840	0.83	
MCC	0.645	0.38	
Sensitivity	0.792	0.48	
Specificity	0.888	0.88	
Note:

* This data was adapted from the SpotOn study.

The top ranking algorithms in the SMOTE and PCA pipeline are used to develop an ensemble classifier to achieve better performance compared to any single algorithm. Different ensemble algorithms are tested: stacking, where a meta-classifier is used to combine the predictive power multiple base classifiers, and voting, a simple ensemble method where each of the six algorithms tested votes on a specific data point, and a simple majority vote is used to predict the classification of that data point as shown in Table 4. In this case, the meta-classifier used during stacking is a Logistic Regression classifier where C = 5. Each individual model is used as a base model separately with the meta-classifier, and all models are combined with the meta-classifier. All ensemble models are run on the SMOTE and PCA pipeline. In the voting ensemble, hard voting was implemented, and all six algorithms are subjected to majority voting. Here, the best performing classifier was the stacking classifier where all models are combined with the meta-classifier. However, the AUROC of this ensemble method is still lower than that of the top individual model, the logistic regression in the SMOTE and PCA pipeline.

Table 4 Different ensemble classifiers (stacking and voting) were tested.

The “AUROC” column is bolded as to highlight the metric used to select the best ensemble classifier. The logistic regression stacked with logistic regression is the best performing pipeline based on AUROC.

	AUROC	Accuracy	Precision	Recall/Sensitivity	F1	Precision-Recall	MCC	Kappa	Specificity	
LR w/Logistic Regression	0.819	0.789	0.525	0.875	0.656	0.488	0.552	0.517	0.763	
RF w/Logistic Regression	0.729	0.740	0.460	0.708	0.557	0.393	0.403	0.385	0.750	
GBC w/Logistic Regression	0.773	0.808	0.567	0.708	0.630	0.469	0.508	0.502	0.838	
KNN w/Logistic Regression	0.656	0.673	0.375	0.625	0.469	0.321	0.271	0.253	0.688	
MLP w/Logistic Regression	0.798	0.779	0.513	0.833	0.635	0.466	0.519	0.489	0.763	
Gaussian w/Logistic Regression	0.581	0.490	0.277	0.750	0.405	0.265	0.141	0.102	0.413	
All (Stacking) w/Logistic Regression	0.785	0.827	0.607	0.708	0.654	0.497	0.542	0.539	0.863	
Voting Classifier	0.710	0.712	0.425	0.708	0.531	0.368	0.365	0.341	0.713	

A comparison of the accuracy and performance of the model developed herein, shown in bold, compared with SpotOn as shown in Table 5. In our model, the logistic regression was our top performing algorithm, and was thus used to develop to predict hotspots with high accuracies. The SpotOn study (Moreira et al., 2017) was used in order to identify the testing accuracies of the SpotOn study and those of the other studies as well. The other studies that are compared to are SpotOn, SBHD213, Robetta23, KFC2-A24, KFC2-B, and CPORT25 (Kim, Chivian & Baker, 2004; Martins et al., 2014; De Vries & Bonvin, 2011; Zhu & Mitchell, 2011).

Table 5 Comparison of our study to other studies.

The “Our model” column is bolded as to highlight the results of our model to other models.

	Our model	SpotOn*	SBHD2*	Robetta*	KFC2-A*	KFC2-B*	CPORT*	
AUROC	0.842	0.91	0.69	0.62	0.66	0.67	0.54	
Sensitivity	0.833	0.98	0.7	0.29	0.53	0.28	0.54	
Specificity	0.850	0.84	0.71	0.88	0.81	0.96	0.47	
F1-score	0.714	0.96	0.62	0.39	0.56	0.42	0.42	
Note:

* Columns 2 through 7 are adapted from the SpotOn study to perform the side-by-side comparison among the algorithms.

Phase 2

The top logistic regression algorithm from the SMOTE and PCA pipeline was utilized to predict the hotspots from the EphB2-ephrinB2 complex. This resulted in consecutive residues 1122–1126 predicted as viable hotspots. Drug screening performed on these hotspots results in nine potential small molecules that could bind to this hotspot. The pharmacophore fit scores are provided from our drug screening analysis, where the highest ranked drug also has the highest pharmacophore fit score as shown in Table 6. Extensive literature review was conducted to identify which drugs could aid in treating conditions associated with the EphB2-ephrinB2 complex.

Table 6 Pharmacophore fit rankings from drug screening: details for top ranked drugs.

Generic name	Pharmacophore fit score	Database ID	Molecular weight	
Pralatrexate	47.41	DB06813	477.47	
Chlortetracycline	46.02	DB09093	478.88	
Nadolol	45.97	DB01203	309.4	
Imipenem	45.51	DB01598	299.35	
Idarubicin	45.46	DB01177	497.49	
Valganciclovir	44.76	DB01610	354.36	
Conivaptan	43.92	DB00872	498.57	
Cimetidine	43.86	DB00501	252.34	
Barnidipine	43.53	DB09227	491.54	

Phase 1

As the best performing classifier, the logistic regression from the SMOTE and PCA pipeline is subjected to PCA, it is difficult to analyze exactly which features are the most valuable. As a result, in order to identify the most relevant features, sensitivity analysis using Python library Pytolemaic (https://pypi.org/project/pytolemaic/) was performed on the best performing logistic regression classifier from the SMOTE-only pipeline to understand the significance of adding new features to the existing dataset as provided by the SpotOn study. Three out of the top ten features (Relative Complex ASA, Complex ASA, and the engineered features Dist-4.0 * Pair Potential) as identified by Fig. 1 were added in this study exclusively, and highlights the improvement in predictive capabilities of the addition of these features.

Figure 1 Feature importances of the top logistic regression classifier.

Sensitivity analysis of the logistic regression from the SMOTE-only pipeline were performed. Features near the top of the graph have higher feature importances.

PyMol (Schrödinger, 2015) was used to derive and highlight residues 1122–1126 on chain E of the EphB2-ephrinB2 complex. Predicted druggable hotspot residues are shown as more visible surface markers (in green), and the other residues are shown in pink or light red in Fig. 2. Residues 1122–1126 were selected for drug screening as consecutive residues may be used as initial fragments in drug screening (Modell, Blosser & Arora, 2016).

Figure 2 The EphB2-ephrinB2 complex with highlighted residues using PyMol.

Residues 1122–1126 are highlighted as shown in green as surface markers. The rest of the complex is in pink.

Phase 2

An apo-site grid, as shown in Fig. 3, was developed and implemented on hotspot residues 1122–1126 as identified via the machine learning model on the EphB2-ephrinB2 complex, where a visual representation of the hotspots on the complex are shown in Fig. 2. This grid was developed by first calculating the pockets of hotspot residues 1122–1126 on LigandScout (Wolber & Langer, 2005).

Figure 3 Apo-Site Grid for residues 1122–1126.

Apo site pharmacophore of residues 1122–1126. The gray parts of the grid indicate the levels of buriedness and surface area.

Figure 4 shows the 26-feature pharmacophore developed using an apo-site grid derived using hotspot residues 1122, 1123, 1124, 1125, and 1126 identified in Fig. 3 via the machine learning model. A pharmacophore identifies the key parts of the molecular features that define the function and shape of a specific ligand, and includes features such as H-bond acceptors and donors, hydrophobic and aromatic rings, etc. This pharmacophore is then used to identify drugs that fit its features. The scoring of this screening procedure follows a pharmacophore-fit scoring function as provided in LigandScout. A maximum number of two features are omitted from this multi-feature pharmacophore to identify small molecule hits, and the best matching conformation is selected.

Figure 4 Pharmacophore model of residues 1122–1126.

Cimetidine, currently an acid reflux medication, was identified via virtual screening to potentially bind to the EphB2-ephrinB2 complex associated with cancer cells as shown in Fig. 5. The right image is cimetidine in relation to the 26-feature pharmacophore developed as shown in Fig. 4. A pharmacophore-fit score of 43.86 was achieved during drug screening. Further literature review identified cimetidine as a potential repositioning target for many different types of cancers, including melanoma, gastric, and colorectal cancers (Pantziarka et al., 2014).

Figure 5 (A) Structure of cimetidine. (B) Relative structure of cimetidine in relation to the developed pharmacophore.

Idarubicin, a chemotherapy medication that’s currently used to treat breast cancer, was identified via virtual screening to potentially bind to the EphB2-ephrinB2 complex, where the expression of the complex is associated with cancer cells as shown in Fig. 6. The pharmacophore fit score of this small molecule is 45.46. This drug was also found to treat cancers linked to the EphB2-ephrinB2 complex such as melanoma and leukemia (Martoni et al., 1986; Jabbour et al., 2017). The right image is idarubicin in relation to the pharmacophore developed as shown in Fig. 4.

Figure 6 (A) Structure of idarubicin. (B) Relative structure of idarubicin in relation to the developed pharmacophore.

Pralatrexate, a T-cell lymphoma medication, was identified via virtual screening to potentially bind to the EphB2-ephrinB2 complex, where the expression of the complex is associated with cancer cells as shown in Fig. 7. This small molecule has a pharmacophore fit score of 47.41, and literature review suggests that this drug could potentially treat breast cancer and prostate cancer (Yu, Zhao & Gao, 2018; Serova et al., 2011). Recent research has shown that pralatrexate can treat esophagogastric cancer, which is associated with the various gastrointestinal cancers of the EphB2-ephrinB2 complex (Malhotra et al., 2020). The right image is pralatrexate in relation to the pharmacophore developed as shown in Fig. 4.

Figure 7 (A) Structure of pralatrexate. (B) Relative structure of pralatrexate in relation to the developed pharmacophore.

Nadolol, a beta blocker, was identified via virtual screening to potentially bind to the EphB2-ephrinB2 complex, where the expression of the complex is associated with cancer cells as shown in Fig. 8. This small molecule has a pharmacophore fit score of 45.97, and literature review suggests that beta blockers could potentially treat a variety of cancers, including breast cancer and pancreatic cancer (Ishida et al., 2016). A close relative of this drug, propranolol, can induce apoptosis in liver cancer cells (Wang et al., 2018). This research suggests nadolol’s potential role in mitigating the effects of other cancers as well. The right image is nadolol in relation to the pharmacophore developed as shown in Fig. 4.

Figure 8 (A) Structure of nadolol. (B) Relative structure of nadolol in relation to the developed pharmacophore.

Virtual drug screening identified nine drugs (pralatrexate, chlortetracycline, nadolol, imipenem, idarubicin, valganciclovir, conivaptan, cimetidine, and barnidipine) that bind to the pharmacophore shown in Fig. 4. Further analysis via literature review identified four drug candidates to potentially treat various types of cancers: cimetidine, idarubicin, pralatrexate, and nadolol. Figure 5 shows the possibility for cimetidine, an antacid, to bind with the EphB2-ephrinB2 complex, and scientific literature identified the possibility for this drug to potentially treat melanoma, gastric, and colorectal cancers (Pantziarka et al., 2014). Figure 6 identifies the possibility for idarubicin, a chemotherapy drug used to treat leukemia, to bind with the EphB2-ephrinB2 complex, and literature review identified the possibility for this drug to potentially treat melanoma and leukemia (Martoni et al., 1986; Jabbour et al., 2017). Figure 7 demonstrates the possibility for pralatrexate, a T-cell lymphoma medication to bind to the EphB2-ephrinB2 complex.

Discussion

In this article, we presented our development of a machine learning approach for identifying druggable hotspots at protein–protein interfaces. Our algorithm builds on previously existing methods, most notably the SpotOn study. Our approach combines molecular features that have not previously been combined, such as the molecular descriptors used in the SpotOn and HotPoint studies, and additional information related to amino acid composition as provided by the protr module. It applies various machine learning techniques, such as 10-fold cross-validation, feature engineering, and ensembling techniques, including voting and stacking. A logistic regression with C = 1,000 was used in order to achieve an AUROC of 0.842, sensitivity/recall of 0.833, and specificity of 0.850.

In order to find the most optimal pipeline, all four pipelines were run, and the pipeline that used SMOTE and PCA during the pre-processing step was chosen the most optimal pipeline due to its high average AUROC score. The average metrics of all classifiers in each of the pre-processing steps are recorded in Table 1. Furthermore, the results of each top performing classifier in the SMOTE and PCA pre-processing step are illustrated in Table 2. In Table 3, the top metrics of the highest performing algorithm in the most optimal pipeline, the SMOTE and PCA pre-processing pipeline, are compared with the top-performing model in the ScaledUp pre-processing dataset in SpotOn, the highest performing dataset in that study. SpotOn-specific metrics are provided by the study itself. The individual models of our study performed better than the individual models of SpotOn as highlighted in Table 3. After this step, ensemble methods such as stacking and voting were implemented to potentially achieve even better results than any single model. The results of performing this step are shown in Table 4.

Although our models outperform that of SpotOn’s individual models on all metrics, the results of our approach are lower on three out of four metrics than the top performing ensemble model from the SpotOn study, as illustrated in Table 5. This may be due to one of many reasons. Even though there was an increase in the total number of features as compared to the SpotOn study, the slight decrease in the total number of samples could potentially negatively affect predictive performance. Another reason could be that the models tested are not diverse enough from each other to significantly boost performance via ensembling. Two of the models in this study are tree-based methods (random forest and gradient boosting). This seems to be the most plausible explanation for why our top individual model outperformed SpotOn’s top individual model on all metrics, but ensembling techniques did not improve performance. A greater diversity of these models would probably have boosted performance during stacking or voting, as a greater variety of base models have been shown to boost predictive performance (Whalen & Pandey, 2013).

To illustrate our approach, we applied this model to analyze the EphB2-ephrinB2 complex, which has been overexpressed and associated with multiple types of cancer, including prostate, gastric, colorectal and melanoma cancers (Pasquale, 2010). As the overexpression of the EphB2-ephrinB2 complex is associated with these cancers, further analysis for drug discovery could aid in identifying possible new hotspots that potentially aid in drug discovery in the fight against cancer (Barquilla & Pasquale, 2015). In addition, the viability for the EphB2-ephrinB2 complex, and more specifically the EphB2 receptor, for drug discovery has been examined, and it was determined that small molecules could potentially disrupt and/or bind to the ephrin binding pocket (Chrencik et al., 2007; Noberini, Lamberto & Pasquale, 2012).

The effectiveness of introducing new and engineered features was demonstrated by the sensitivity analysis on the logistic regression on the SMOTE-only pipeline (Fig. 1), where three out of the top ten features were added in this study exclusively. Our algorithm identified a set of residue hotspots (Fig. 2), which were then used to generate an apo-site grid and pharmacophore model (Figs. 3 and 4). This model was used to identify drugs with similar characteristics that could be potentially used to modulate the molecular functions of the EphB2-ephrinB2 complex. Table 6 outlines the nine small molecules that passed the drug screening test. Extensive literature review was performed on all nine drugs, and four small molecules were selected their potential efficacy regarding their ability to treat conditions associated with the EphB2-ephrinB2 complex. The identified drugs included compounds already used for cancer treatment, such as pralatrexate, a T-cell lymphoma medication, as well as non-cancer medication; cimetidine, an antacid; and nadolol, a beta blocker that can treat cardiac conditions. Literature review suggests that pralatrexate can potentially treat breast cancer and prostate cancer, and recent literature highlights the possibility for this small molecule to treat other conditions such as cancers of the gastric and esophageal systems (Yu, Zhao & Gao, 2018; Serova et al., 2011; Malhotra et al., 2020). Figure 8 identifies nadolol, a beta blocker that can treat cardiac conditions, as a candidate to bind to the EphB2-ephrinB2 complex. Literature review strongly supports that beta blockers can be repositioned to treat other cancers, such as cancer, and has identified a close relative of nadolol, propranolol, as a potential treatment against multiple cancers, including colon cancer (Işeri et al., 2014).

Conclusion

The model developed herein in phase one compares favorably with those developed in prior studies and offers enhanced predictive ability for identifying new druggable hotspots, including possible druggable hotspots for cancer-related protein interfaces. The predictive capabilities of the model developed herein are high, offering a high AUROC and overall predictive performance to date. Herein, a logistic regression with C = 1,000 was utilized to successfully identify hotspots.

Phase two of this project aims to identify possible drugs for repositioning. Structural properties of the identified hotspot residues, such as H-bond acceptors and donors, were identified as feature sets to aid in drug development. The efficacy of the model developed herein has been demonstrated through its successful ability to predict drug-disease associations previously identified in literature, including cimetidine, idarubicin, and pralatrexate. Importantly, nadolol has been uniquely identified in this study to potentially treat conditions caused by the overexpression of the EphB2-ephrinB2 complex. This work aims to yield better predictions in terms of hotspot discovery by primarily increasing the sheer amount of data that is available regarding protein–protein interactions. As a consequence, this work has shown that the increases in predictive power as a result of this addition of data.

Possible avenues for future work include drug development using the pharmacophores identified in this study to treat these diseases. Hopefully, by identifying hotspot residues with unparalleled accuracy and identifying possible drug repositioning opportunities, traditional drug development based on these residues and repositioned drugs could yield new and effective treatments for diseases such as cancer. In addition, adding additional novel features and data for hotspot identification, especially those that directly correlate with the extent of how energetically favorable residues are, could further improve model performance. Another avenue for future work would be to streamline the workflow of both phases. Phase one is automated with the help of the machine learning model. However, phase two requires manual input of the hotspot residues as identified in phase one to identify potential drug candidates. A more streamlined process would improve functionality and ease of use.

Supplemental Information

Supplemental Information 1 The training metrics for the top performing algorithms.

The individual training and testing metrics of each of algorithms developed and tested per pipeline, and during different ensembling techniques.

Click here for additional data file.

Supplemental Information 2 Dataset used for Phase 1.

Raw data used to develop the machine learning model. This is built on top of the dataset from the SpotOn study. Features were added from the protr module, the HotPoint study, and through manual feature engineering.

Click here for additional data file.

Supplemental Information 3 Code and Python methods, and the whole dataset.

This code works standing alone.

Click here for additional data file.

I would like to thank the researchers who conducted the SpotOn study, especially Ms. Irina Moreira, for providing the code and existing dataset that this study is built on top of. I would also like to thank Dr. Michael McKelvy of Basha High School for his extensive feedback on my poster and project. In addition, I would like to thank Mr. Thomas Lemker for his assistance in using the LigandScout software.

Additional Information and Declarations

Competing Interests

Author Contributions

Data Availability

The authors declare that they have no competing interests.

Rohit Nandakumar conceived and designed the experiments, performed the experiments, analyzed the data, prepared figures and/or tables, authored or reviewed drafts of the paper, and approved the final draft.

Valentin Dinu analyzed the data, authored or reviewed drafts of the paper, and approved the final draft.

The following information was supplied regarding data availability:

All data and code used to produce these results are available as Supplemental Files.

The code for the SpotOn study was obtained from Dr. Irina Moreira and her lab (irina.moreira@cnc.uc.pt) and the data from the HotPoint study is available at http://prism.ccbb.ku.edu.tr/hotpoint.

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
