# Peer review of "Developing a machine learning model to identify protein–protein interaction hotspots to facilitate drug discovery"

_PeerJ, doi:10.7717/peerj.10381_

## Round 0.1 · original submission · Major Revisions

While the reviewers were generally positive, a number of concerns were raised that should be addressed in the resubmission.

Reviewer 1 ·

Basic reporting

It is fine. No comment.

Experimental design

It is fine. No comment.

Validity of the findings

There are many many algorithms and machine learning approaches to predict protein-protein interaction, hot spot, drug binding affinity, etc. Most of these methods differ slightly in various metrics. The new features added into this study ( features related to amino acid composition, dipeptide composition, etc., to the already pre-existing data) should be listed in a Table. Also in the final results as in Figure 1, are the new features in the top list, and how the adding of new features change the overall distributions of feature importance.

The second question is the identification of the potential medications listed in the paper. Figures 5-8 should be combined into a larger figure with four panels. More description about the identification and selection should be provided. Details for top screened drugs is better in main text, with additional entries to prove that these drugs are outstanding from virtual screen. What are rankings of these drugs among hits. Otherwise, it is dubious that these drugs could be hand picked.

Additional comments

Overall, it is important to improve accuracy of machine learning in predicting various protein-protein interaction. This paper represent a rigorous approach to add more relevant features to screen hot spot related factors.

Reviewer 2 ·

Basic reporting

See 'general comments'

Experimental design

See 'general comments'

Validity of the findings

See 'general comments'

Additional comments

The authors presented a method for predicting protein-protein interface hotspots using supervised machine learning algorithms trained using an extensive number of sequence and structure derived features. The viability of the developed method has been demonstrated using a virtual drug screening analysis of the predicted hotspots on the EphB2-ephrinB2 complex.

Dataset: The final dataset is slightly different from the original SpotOn dataset. Therefore, the authors need to report the number of hotspot and non-hotspot residues in their dataset.

Performance evaluation: Since the dataset is imbalanced and have less number of samples compared with the number of extracted features, the estimated performance metrics might be sensitive to the portioning of the data into train and test sets. To address this issue and have a fair comparison with SpotOn, the authors need to evaluate their models using 10 runs of 10-fold CV experiments.

Table 3: The authors claim that their model outperforms SpotOn (based on MCC, F1, and sensitivity) is not accurate. First, SpotOn has AUC score of 0.83 while the authors model has an AUC score on only 0.75. Second, while the authors model has a better sensitivity, its specificity is 0.08 lower than SpotOn. Third, MCC and F1 tend to be higher for the model with the highest specificity. Fourth, the authors model has been evaluated using a single run of 10-fold cv while SpotOn model has been evaluated using 10 runs. Ideally, the two models need to be compared by inspecting the ROC curves. Since it is challenging to get the ROC curve for SpotOn, the authors might modify the threshold of their model to achieve an 0.88 specificity and show that at this specificity their model has a better sensitivity and hence better F1 and MCC metrics.

Feature importance: Lines 248-251 showed that Relative Complex ASA and Complex ASA features are among the top 15 features. These two features will assign the same value to hotspot and non-hotspot residue in the complex. If this is the case, how can these features have any discriminative signal? Moreover, to demonstrate the improvement in the predictive performance by the newly added features, a sensitivity analysis to quantify the change in the model performance before and after adding these novel features is needed.

---

## Round 0.2 · accepted · Accept

Thank you for addressing the reviewers' concerns and congratulations again.

Reviewer 1 ·

Basic reporting

ok

Experimental design

ok

Validity of the findings

ok

Additional comments

No more comment. Thanks for revision